# Climate warming has compounded plant responses to habitat conversion in northern Europe

Alistair G. Auffret [1] ✉ & Jens-Christian Svenning [2]

Serious concerns exist about potentially reinforcing negative effects of climate change and land conversion on biodiversity. Here, we investigate the tandem and interacting roles of climate warming and land-use change as predictors of shifts in the regional distributions of 1701 plant species in Sweden over 60 years. We show that species associated with warmer climates have increased, while grassland specialists have declined. Our results also support the hypothesis that climate warming and vegetation densification through grazing abandonment have synergistic effects on species distribution change. Local extinctions were related to high levels of warming but were reduced by grassland retention. In contrast, colonisations occurred more often in areas experiencing high levels of both climate and land-use change. Strong temperature increases were experienced by species across their ranges, indicating time lags in expected warming-related local extinctions. Our results highlight that the conservation of threatened species relies on both reduced greenhouse gas emissions and the retention and restoration of valuable habitat.

Climate and land-use change are widely accepted as the principal drivers of biodiversity change in the Anthropocene[1–3], and legitimate concerns exist regarding their potentially reinforcing negative effects on populations and communities[4,5]. An important concern is that species responding to climate change need suitable habitat in which to persist and spread. However, climate change can also accelerate land-use change[6,7], while the type of land-use change that occurs in any given location also affects the level of warming that organisms experience on the ground[8]. Despite these pressing issues, data limitations regarding land use, climate and species occurrences mean that there is still only scant evidence regarding how these two threats interact to drive biodiversity change, especially over the large spatial and temporal scales relevant for studying changes in species distributions.

Here, we use plant occurrences from regional plant atlas inventories in four provinces of Sweden to analyse the dual effects of climate warming and habitat conversion on the regional distributions of 1701 vascular plant species over a period of approximately 50–80 years

(Fig. 1; 'Methods'). Analysis of historical maps has revealed that during the same time period, these regions have experienced a widespread abandonment of species-rich grassland habitat and subsequent afforestation, while temperatures have warmed by almost 1.5 °C[9,10]. We investigate how changes in climate and land use have affected plant species' regional distributions, with an emphasis on uncovering any tandem or interactive effects of the two drivers. First, we quantify species' distribution changes in relation to their habitat and climatic associations, both in general and in terms of directional climate-related shifts, finding relatively higher declines in grassland specialists and expansions in warm-climate species. Then, we study how changes in the environmental conditions in the landscape affect the level of community turnover that has occurred over time, with results indicating that retention of grassland habitat can prevent extirpations, while combined grassland abandonment and climate warming facilitate colonisation. Despite both high variation across species and low explanatory power of models indicating additional drivers of change, our correlative analysis indicates that compounding effects of habitat

[1]Department of Ecology, Swedish University of Agricultural Sciences, Box 7044, SE-75 007 Uppsala, Sweden. [2]Center for Biodiversity Dynamics in a Changing World (BIOCHANGE) & Section for Ecoinformatics and Biodiversity, Department of Biology, Aarhus University, Ny Munkegade 114, DK-8000 Aarhus C, Denmark. ✉e-mail: alistair.auffret@slu.se

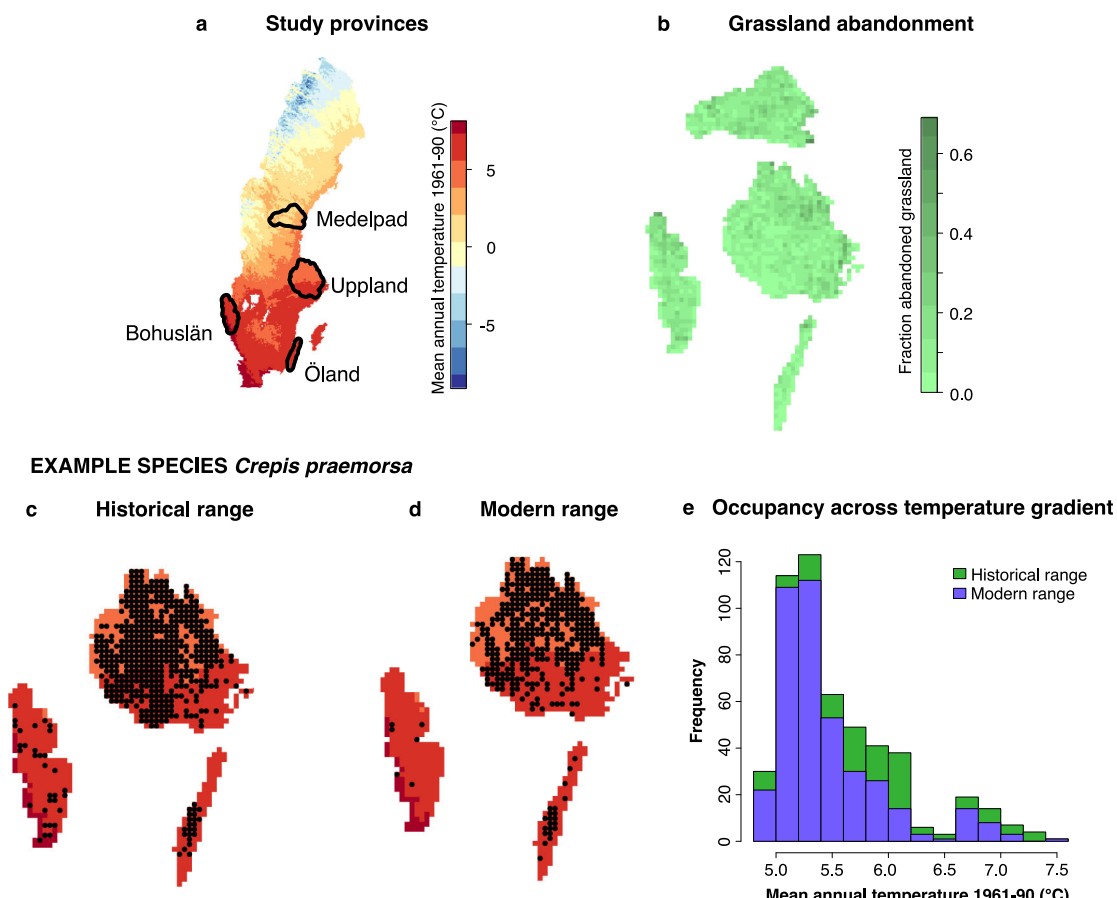

**Fig. 1 | Study system. a** Data in the study are taken from four Swedish provinces along a climate gradient. **b** Since the 1940s–60s, large amounts of grassland have been abandoned in all provinces, with high fractions of 5 × 5 km grid cells now covered with forest that was on previously open land. Observation data were taken from historical and modern plant atlases, translated into occupancy in 5 × 5 km grid cells. Panels **c** and **d** show change in observations of red-listed (NT) *Crepis praemorsa* in three provinces where distributions were mapped, indicating an overall distribution loss. **e** Modern observations appear to be especially fewer in the southerly, warmer half of its distribution.

conversion and climate warming may lie behind observed changes in local patterns of species occurrences and regional distributions.

## Results

### Correlates of regional distribution change

To examine which types of species are experiencing regional distribution changes (defined by the presence in grid cells of 5 × 5 km), we first calculated for each species within each province a metric of distribution change using the Frescalo algorithm[11] (Supplementary Data 1). Frescalo uses a combination of geographic proximity and overall floristic similarity within local neighbourhoods to determine spatial variation in observer effort. This is used together with observed occurrences of common benchmark species to estimate the probability of occurrence of each species within each focal grid cell at each time period. These probabilities are used to calculate estimates of relative frequency for each species, which are then compared across time periods to calculate a metric of distribution change. This approach is recommended for data–like ours–that have been collected across discrete time periods and with an unknown observer effort[12]. We then regressed distribution change metrics for each species in each province against species traits regarding climatic and habitat associations using a linear mixed model. Grassland and forest specialisation were defined as the proportion (scale 0–10) of each species' national population that are found within these broad habitat types[13]. For climate associations, species mean temperature index describes the mean temperature value of recorded occurrences across their range in Sweden, based on the climate conditions at the time and

location of 7.3 million observations from provincial floras published since 1975 across the country[10]. Larger values indicate warmer, more southerly distributions and lower values cooler, northerly distributions. Species temperature range index is the difference between the warmest and coolest areas of the Swedish range, with higher values indicating more climatically widespread species. Our model showed that species with relatively warmer climate associations were more likely to expand their regional distributions, while grassland specialist species are more likely to have declined (Fig. 2a, b, Supplementary Data 2). Climatically widespread species also showed relatively positive distribution trends, although increases were less strong in more forest-specialised species, as evidenced by a significant negative interaction term (Fig. 2c). These results reflect the overall trends of the environmental changes that have occurred in Sweden and elsewhere[9,14,15], and extend the observation of local-scale extirpation of grassland species[14,16] to an overall decline in their broad-scale regional distributions. Although many species are declining (negative distribution trends), forest species are more likely to have increased their ranges. However, the negative interaction with climatic distribution size indicates that widespread forest species of early secondary successional habitats are more likely to have benefitted from recent environmental change, rather than species that are specialised to mature forest conditions[17].

### Shifts in climatic space

To assess the direction of regional distribution gains and losses in terms of climatic ranges, a second set of analyses compared how

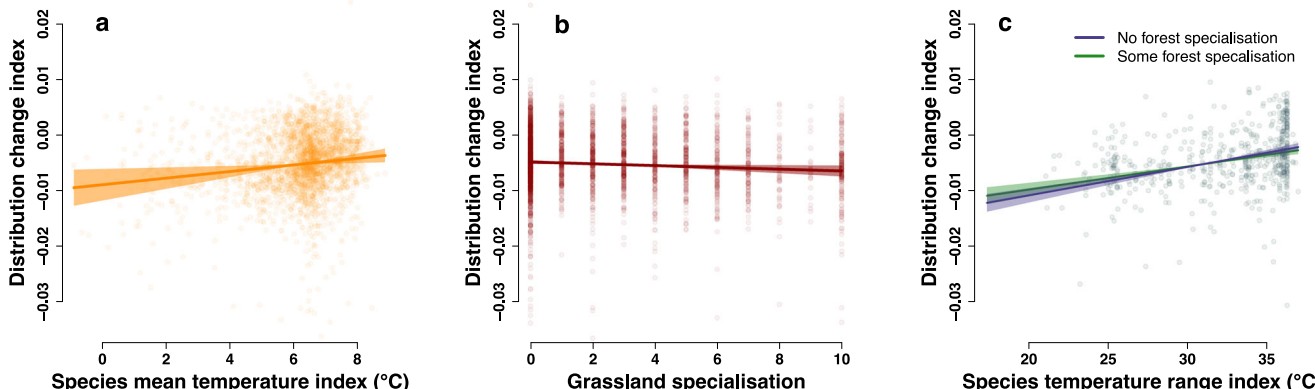

**Fig. 2 | Correlates of regional distribution changes in plant species over time.** Points and lines show the relationships between measures of **a** species mean temperature index (the average temperature experienced by a species across its Swedish range[10]), **b** grassland specialisation (the proportion of a species' population in grassland habitats[13]) and **c** species temperature range index (the range of temperatures experienced by a species across its Swedish range[10]) with Frescalo distribution change index[11], based on a linear mixed effects model (*n* = 2771 estimates of change from 1431 species in the four provinces). In **c**, the effect of species temperature range is split according to species with no and slightly higher forest specialisation values (the proportion of a species' population in forest habitats[13]; 0, 2). Ribbons show 95% confidence intervals around the predicted relationship. Y-axis is truncated, excluding 15 species with distribution change index < −0.0374 and 1 species with distribution change index >0.0274. Model R²-marginal = 0.04, R²-conditional = 0.2). See Supplementary Data 2 for full model outputs.

species' regional distribution changes from the historical to the modern period have resulted in shifts in the climate space that they occupy. As the climate warms, species are expected to follow their climatic niches, shifting their ranges to higher latitudes[18,19]. However, because our regions are not contiguous in space to the extent that latitudinal shifts can be calculated, we use a climatic space approach. Here, we used records from a subset of well-recorded cells, defined as those that contain observations of at least 10% of the province's observed species and at least 25% of the species found in the eight adjacent grid cells. Such an approach has previously been used to ensure as far as possible that observed absences are equivalent to true absences when estimating climate-driven range shifts[20,21]. We estimated shifts in climatic space in two ways (Supplementary Data 3). First, grid cells were assigned values for average mean temperature during the 1961–1990 reference period, and the temperature across occupied grid cells was calculated for each species for both the historical and modern time period. Average temperature values from the historically-occupied grid cells were then subtracted from temperature values from the modern occupied grid cells, which resulted in an estimation of geographic shifts across a spatial temperature gradient. That is, the warming-driven expansion of a species that is following its climatic niche to higher latitudes would result in a shift to grid cells that have a lower (cooler) reference period temperature. For our second approach, the average temperature across occupied grid cells in the historical period were calculated for each species according to historical temperatures (1961–1970), while the average temperature across occupied grid cells in the modern period were calculated according to modern temperatures (2001–2010). In this case, by subtracting the historical values from the modern values, we calculated shift in climate experienced by a species across its range over time. As such, a species retaining exactly the same range (in terms of occupied grid cells) would simply experience a shift in climate space equal to the average change in temperature that has occurred in those grid cells.

We found that despite substantial variation, species were generally more likely to shift in geographic space towards cooler, rather than warmer climates in terms of reference-period temperature (Fig. 3a; paired, one-sided Wilcoxon test *p* < 0.001). In other words, species appeared to be exhibiting range changes in the direction expected following climate warming. On the other hand, the increases in temperature that have occurred across the study regions mean that 99% of studied species are now experiencing warmer temperatures

across their ranges than previously (Fig. 3a). This highlights that most species are probably exhibiting disequilibrium responses to recent climate change, indicating potential lags in immigration or extirpation[22,23].

To investigate if shifts in climatic space are related to species-level habitat or climate associations, we used a linear model to assess if these associations could explain the observed shifts in climate space along the spatial (reference-period) temperature gradient, i.e. where negative values in the response variable indicate a climate-driven shift to relatively (in space) cooler areas, and vice versa. Results indicate that warm-climate, southerly species are more likely to have shifted to ranges with on average higher reference-period temperatures, while the opposite dynamic is seen for cold-climate, northerly species (positive relationship between climate space shift and species mean temperature index; Fig. 3b). Although this might initially appear counter-intuitive (one might expect warm-associated species to be the species that are expanding to higher latitudes, i.e. to grid cells with lower reference-period temperatures), it can be explained in two ways. First, warmer-associated species are consolidating, or filling their existing (relatively warmer) range to a greater extent than they are expanding to cooler areas[24], although this effect was smaller for forest specialists (Supplementary Data 4). Second, relatively cold-climate species can over time have shifted to occupy grid squares that are cooler on average (in terms of reference period temperatures), as they are gradually extirpated from the (relatively) warmer areas of their ranges[25,26]. Cold-climate species have also been demonstrated to be particularly susceptible to climate warming in experimental settings[27,28]. Such extirpations could be due to temperatures becoming unsuitably high, but the pattern might also be exhibited by cold-climate species declining for other reasons that result in a retraction to the core (cool) area of the range. Indeed, we also found that species with higher grassland specialisation were more likely to occupy grid cells with cooler reference-period temperatures over time. Remembering from the first analysis that regional distributions of grassland species are declining as a whole, faster declines in relatively warmer areas can potentially be explained by likely faster levels of secondary succession following grassland abandonment[7,29]. This is supported by a significant interaction term, showing that southerly-distributed grassland specialists are 'shifting' to cooler ranges at a faster rate (Fig. 3c). It is also possible that there are other reasons for losses of grassland specialists in

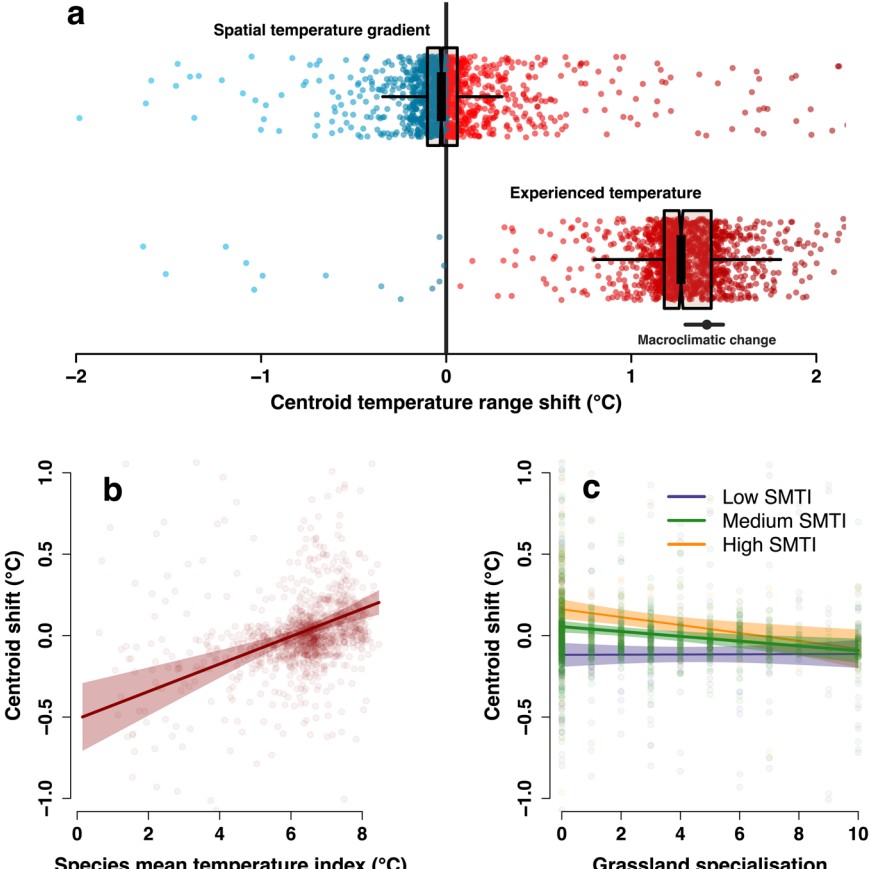

**Fig. 3 | Species' regional distribution shifts in climate space.** Boxplots show shifts in the average climate experienced across a species' range, by subtracting mean temperature values of the historical range from those of the modern range ($n = 1391$ species). Shifts are shown both **a** in terms of a spatial temperature gradient (based on reference period temperature 1961–90) and experienced temperature (1961–70 in historical period, 2001–2010 in modern period). Boxes show median and interquartile range, with whiskers indicate range excluding outliers. Notches represent 95% intervals around the median. Blue and red points show species with shift values below and above zero, respectively. Black point and whiskers indicate the median and interquartile range of climate change experienced across all grid cells. Lower panels show the effect of **b** species mean temperature index (SMTI) and

**c** grassland specialisation on the shift in climatic space for each species (generalised linear model; $n = 1296$ species) according to reference period temperatures, whereby a negative shift indicates a climatic shift to relatively cooler, more northerly climates. In **c**, effect of grassland specialisation is split according to low, medium and high values of SMTI (4.67, 6.54, 7.69). Lines show modelled relationships with ribbons showing 95% confidence intervals. Y-axes are truncated, in **a** excluding 16 species with spatial gradient centroid shifts above 2.16 °C and below −2.16 °C, and 49 species with experienced temperature centroid shifts outside of that range; in **b** and **c**, 47 species have predicted values above 1.08 °C and below −1.08 °C. Model $R^2 = 0.05$. See Supplementary Data 4 for full model outputs.

more southerly areas of Sweden, for example because those regions are broadly more agriculturally-intensive.

## Local drivers of turnover

In a third and final set of analyses, we investigated the extent to which measured changes in temperature and land use in a grid-cell have affected the turnover of species within the grid cell over time (Supplementary Data 5). We created two suites of generalised linear mixed models to establish how climate and land use explain [1] the fraction of species extirpated from a grid cell and [2] the fraction of the province's species pool that colonised a grid cell. Separate models were built using predictor variables relating to the baseline historical conditions (grassland habitat, temperature and their interaction), modern conditions (retained grassland habitat, temperature and their interaction) and the change in conditions that have occurred over time (grassland abandonment, temperature change and their interaction). Each model also included controls for observer effort, spatial autocorrelation, microclimate (which can affect species responses to macro-climatic change[8,30]) and latitude[31].

Results show an overall dominance of temperature in driving turnover (Supplementary Fig. 1 and Supplementary Data 6). More extirpations occurred in warmer areas, while the opposite was true for

colonisations, which were concentrated in cooler areas, indicating warming-driven range change. On the other hand, the historical presence, and the retention of grassland habitat, resulted in fewer extirpations overall. Importantly, model interaction terms provided evidence indicating compounding effects of climate and land-use change on species turnover through interactive effects. First, warming was found to reduce the effect of grassland habitat in preventing extirpations, with a positive interaction in the modern model indicating that warmer temperatures lower the extirpation-reducing effect of grassland retention (Fig. 4a). Second, grassland abandonment reinforced the positive effect of warming temperatures on colonisation (positive interaction in colonisation change model; Fig. 4b). Further analyses concentrating on ecologically-relevant subgroups of species provided more evidence of interacting effects of climate and land use. Warm-climate species were shown to colonise cells containing relatively more grassland (both historically and in the modern period), while forest species were more likely to increase in cells with higher levels of climate warming (Supplementary Fig. 1, Supplementary Data 6).

## Discussion

Our three levels of analysis demonstrate how climate warming and habitat conversion are jointly shaping the re-distribution of plant

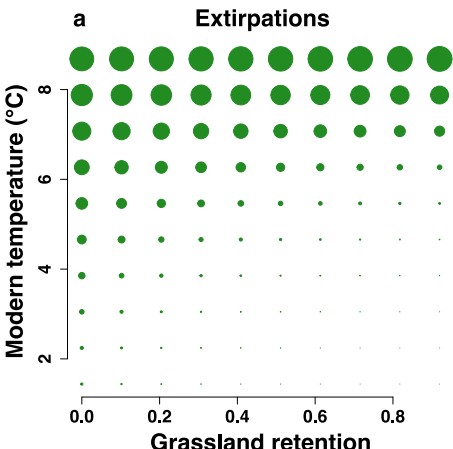
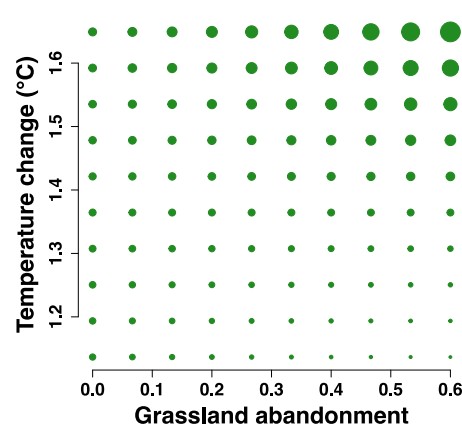

**Fig. 4 | Interactive effects of land use and climate change on plant community turnover.** Panels illustrate effect of land-use–climate interactions on **a** extirpations and **b** colonisations of all species within a grid cell, based on binomial generalised linear mixed effects models ($n = 1231$ grid cells). The size of the points indicates the relative level of extirpation or colonisation in a grid cell over time, according to the values of each predictor variable. In **a**, grassland retention was related to fewer extirpations, but this effect was reduced in warmer climates (model $R^2$-marginal = 0.12, $R^2$-conditional = 0.49). In **b**, high levels of grassland abandonment and temperature increases were associated with the most colonisations (model $R^2$-marginal = 0.04, $R^2$-conditional = 0.16). See Supplementary Data 6 for full model outputs.

species in space and over time in this northern-temperate to boreal region. We show that at our 5 × 5 km 'landscape' scales, warmer temperatures and the large-scale abandonment of grassland habitat has resulted in range contractions in grassland specialist species and species with relatively cool (northern) and narrower climatic ranges. On the other hand, species associated with warmer and more widespread climatic ranges, as well as forest species, are more likely to have expanded their regional distributions.

While a warming climate is known to cause range changes in plant species[32,33], and the cessation of grazing management has led to local losses in grassland specialists[34,35], we show that these two aspects of environmental change are in tandem and interactively driving regional changes in species' regional distributions. Because warmer temperatures are associated with faster woody growth and secondary succession[6,29], it follows that grassland specialists should be particularly negatively affected in warmer areas of their ranges, and that grassland abandonment in combination with warmer and warming temperatures should be associated with higher rates of turnover (Figs. 3 and 4). This compounding negative effect of global change drivers supports the existence of the so-called deadly cocktail of habitat conversion (in this case grassland abandonment, a major driver of biodiversity loss in Europe[9,36,37]) and climate change[4], at least for a subgroup of plant species. On the other hand, our results also show that there are many successful species—notably forest and warm-climate species—that have been able to take advantage of ongoing grassland abandonment in order to shift their ranges in response to a warming climate. Previous studies that have also reported spread of widespread cosmopolitan species in response to anthropogenic pressures over time suggest that such changes lie behind observed patterns of local increases in species richness, but also increased biotic homogenisation[17,38,39].

Habitat conversion is widely regarded as the most important threat to biodiversity worldwide[1,2], and research in European forest ecosystems finds that management-driven changes in the forest canopy can override macro-climatic changes in determining climate-related community shifts[8,40]. In contrast, our study provides evidence that climate might currently be the principal driver of plant community turnover at landscape scales in our cool-temperate and boreal study region. Effect sizes of species traits related to climate were stronger than those related to habitat specialisation, while climate was a stronger and more consistent driver of turnover at the grid-cell level. However, it is important to note that there are other environmental

pressures driving changes in plant species distributions, especially at relatively broad sampling unit of our grid squares. This is evidenced by the relatively low $R^2$-marginal values of our mixed models and $R^2$ of our linear model, indicating substantial variance in species responses that are not explained by our predictor variables. This could for example include changes in land-cover and intensity other than grassland abandonment[41], or nitrogen deposition, which has also affected plant community composition in Europe in recent decades[17,42]. Nonetheless, studies are increasingly identifying climate change as a primary driver of plant diversity change[43,44], but our study is among the first to be able to use land-use data from historical maps to analyse the dual effects of climate together with habitat conversion on long-term floristic change over large spatial scales.

There is a growing body of evidence that plant species respond slowly to environmental change[23,45–47]. This is due in part to plants' various mechanisms for persistence[48,49], as well as a widely-reported limitation in the long-distance dispersal needed to respond to large changes in climate and land use[50,51]. Our results support these trends, as almost all the plant species studied are now occupying warmer ranges today than they were 60 years ago. At the same time, we and others[44,52] show that the environmental changes that have already happened have resulted in substantial changes in the distributions of many species, often in the direction expected according to climatic range shifts. These patterns are consistent with warming-driven, but disequilibrial range responses involving immigration lags as well as extinction debts[22]. According to climate niche theory, species occurrences outside their currently suitable climate are predicted to eventually become extirpated, with consequences for local biodiversity and long-term species survival[53,54]. Therefore, it is possible that despite many grassland and cold-climate species already experiencing range retractions in our study region, there could still be a climate-related extinction debt waiting to be settled[22]. Nonetheless, our results indicate that baseline environmental conditions, modern conditions and environmental change all describe turnover approximately equally-well. Crucially, we show that grassland habitat availability and retention is associated with the persistence of not only grassland specialists, but also cold-climate species and declining species as a whole. Such habitats are still under threat, and together with reductions in greenhouse gas emissions to prevent catastrophic climate change, it is also imperative that we retain these, and other natural and semi-natural habitats in the landscape through the restoration of natural or semi-natural grazing regimes. Restoring large grazers back into landscape

will not only counteract homogenising woody densification[55,56], but is also important for enhancing dispersal and facilitating colonisations to track climate warming in many plant species[57,58].

## Methods

### Occurrence data

Plant occurrences used in this study originated from historical and modern-day regional plant biodiversity atlases (floras) from four Swedish provinces: Bohuslän[59,60], Medelpad[61], Öland[62] and Uppland[63,64]. The historical floras document observations from as early as the 1800s, although the majority of records originate from inventories from the author of each flora during the first half of the 20th century. Georeferenced observation records from the historical floras were derived by the authors of the modern floras in Bohuslän and Medelpad, while in Uppland and Öland they were digitised from distribution maps published in the historical flora[10,65]. Modern observation records were taken from the databases used for the modern inventories, which were concentrated from 1990 to 2020, although in Medelpad, some observations date back to 1975. Species names across all floras were harmonised to the species level (i.e. Genus epithet only), according to the Swedish Taxonomic Database (https://www.dyntaxa.se/; retrieved April 2016), with some species of e.g. Alchemilla, Rubus, Ranunculus and a number of Asteraceae assigned to Section only. Observations were assigned to a 5 × 5 km grid cell. Species within each province were only retained if they were observed in both the historical and modern period, and grid cells that only contained observations from one time period were removed. In total, 1701 species were retained for analysis across 1232 grid cells, 1459 of these species were found in Bohuslän (271 grid cells), 918 in Medelpad (194 grid cells), 286 on Öland (88 grid cells) and 438 in Uppland (679 grid cells). Data for the latter two provinces contain fewer species because georeferenced historical observations were only available for those species that were mapped.

### Climate and habitat associations

Climate indices for each species were taken from Auffret and Thomas[10], where 7.3 million observations of 3053 plant species from provincial floras across Sweden that were published since 1975 were coupled with the mean annual temperature at the location and time of each observation. Species mean temperature index is the mean value of each species' observations, while species temperature range index is the difference between the highest and the lowest temperature experienced across the species' range. Habitat specialisation was calculated using Tyler et al.[13], which estimates the proportion of a species' Swedish population that is found across 38 habitat types on a scale of 0–10. Here, we calculated broad specialisation in grassland habitats by summing the values for dry heath, dry meadow, steppic meadow, moist heath, moist meadow, moist calcareous meadow, poor acidic fen, intermediate rich fen, rich calcareous fen, tall herb-sedge-reed meadow, and seashore meadows. Forest specialisation was defined as the total proportions in heath type deciduous forest, intermediate deciduous forest, herb-rich deciduous forest, heath type coniferous forest, *Vaccinium* type coniferous forest, herb-rich coniferous forest, calcareous conifer forest, subalpine *Betula* forest. Due to these classifications, grassland specialists are generally characterised by species restricted to habitats that require grassland management, while forest specialists are largely represented by species that are found in wooded habitats, and not only so-called ancient forest species.

### Climate and land use data

Temperature data were taken from the Swedish Institute for Hydrology and Meteorology's database *pthbv*, which contains daily modelled climate data from 1961 to present across a 4 × 4 km grid. We aggregated these data to monthly average temperatures, and then resampled the grid so that it matched our 5 × 5 km grid cells of the species

data. We then calculated mean annual temperatures across three time periods: a 1961–1970 historical period, a 2001–2010 modern period and the 1961–1990 reference period. Historical land use was taken from the Swedish economic map, which was created between the 1930s and 1960s in our study area in individual maps across the same 5 × 5 km national grid used in this study. Digitised maps were taken from Auffret et al.[66], which presents land cover in the following categories: arable fields, forest, open areas (mainly grasslands, but also wetlands and urban land uses) and surface water at the 1 m resolution. Modern land-use data were taken from the 10 m resolution national land-cover map *Nationella marktäckedata* from 2018. The 25 land-cover classes were aggregated to match the four broad categories of the historical digitisations, with clear-cut forests classed as forest because it does not represent grassland habitat. For our analysis, we first resampled the historical land use to match the resolution of the modern maps, and then calculated in each grid cell [1] historical grassland habitat: the fraction of pixels that were open land in the historical period, [2] retained grassland habitat: the fraction of pixels that were open land in both the historical and modern period, [3] grassland abandonment: the fraction of pixels that were open land in the historical period and forest in the modern period.

### Observer effort

An issue when comparing species observations over time is that of uneven recorder effort. This can potentially lead to unreliable estimates of species-level change both in terms of regional distributions, but also local patterns of occupancy and turnover. In our dataset, observer effort was higher in the modern period, with individual species-grid cell records (only including species recorded in a province in the historical period) increasing by 1.5% in Uppland, 26% in Bohuslän, 37% on Öland and 432% in Medelpad. Historical plant atlases were largely based on the inventories of one individual, who sought to document the distributions of plant species within the province, usually at the parish level. However, it is rare to find only one record of a species in a parish, and between 40% (Medelpad) and 61% (Öland) of all historical observations were duplicates at the species-grid cell level (grid cells being considerably smaller than parishes). Modern atlases, on the other hand, are generally the work of many individual members of a botanical society, who take responsibility for surveying one or more 5 × 5 km grid cells over a number of years, supported by a core group who may assist in surveying areas across the province. This is likely the main reason for the disparity in observations over time. Nonetheless, both historical and modern plant atlases are made with the specific aim of recording species distributions within a province, and include good representation of both generalist and specialist species recorded in all provinces (Supplementary Data 7). Therefore, we are convinced that although there is a need to correct for uneven observer effort at the grid-cell level, both the historical and modern plant atlases represent reliable sources of data that include surveys of a variety of habitat types across each province, and can therefore be used as a basis for studying the effects of land-use change on plant species.

We used the Frescalo algorithm to both estimate recorder effort for each grid cell, as well as to calculate an index of regional distribution change for each species in each province[11,67]. Briefly, Frescalo defines a neighbourhood for each grid cell based on geographic proximity and total floristic similarity. Then, local frequencies of all species within each grid cell in each neighbourhood are calculated, weighted according to relative proximity and floristic similarity. These are used to produce standardised mean frequencies of species within a grid cell $\varphi_i$, from which a sampling effort multiplier $\alpha_i$ is calculated, which transforms input values of standardised mean frequencies to a target value $\Phi$ representing a more-or-less completely-recorded grid cell. To calculate a metric of distribution change, probabilities of individual occurrence for species in each grid cell and time period are

estimated, based on the value of the sampling effort multiplier and the proportion of common 'benchmark' species ($R^*$) that were recorded there. These are then used to calculate an estimated relative frequency for each species across its distribution per time period, and differences across time periods represent a metric of distribution change. Frescalo has been demonstrated to be the most robust option for estimating distribution change from presence-absence data representing broad time periods, such as plant atlases[12], and is commonly-used by experts to estimate species trends by researchers[68,69], national taxonomic experts[70,71], and for policy documents[72]. We used the Frescalo implementation of the sparta package version 0.2.19[67] in R version 3.4 (version 4.2 was used for all subsequent analyses)[73]. Neighbourhoods were defined by taking the 50 geographically-closest grid cells to a focal cell, and selecting from those the 25 with the most similar vegetation in the modern period (where sampling effort is assumed to be more complete, and prior to removing species that were not recorded in the historical period, see above). We did not set a target mean frequency value $\Phi$, instead allowing the algorithm to select one for each province based on the data, defined as the 98.5th percentile of the observed values of standardised mean frequencies[11]. The proportion of species set as benchmark species ($R^*$, known as 'alpha' in sparta[67]) was 0.1. This is lower than the default value of 0.27, and was chosen because the mapped species in Öland and Uppland did not include the most ubiquitous species in the region (because species with distributions covering the whole province were not considered to require printing a map).

To test for the sensitivity of Frescalo to different values of $R^*$ in the calculation of species-level trends, we also ran the algorithm using $R^*$ values of 0.2 and 0.3. Species trends were highly correlated (Pearson's rho $R^*$ 0.1 and 0.2 = 0.99; $R^*$ 0.1 and 0.3 = 0.98; Supplementary Fig. 2), although the number of species increasing (positive trend values) increased, and the number of species decreasing (negative trend values) decreased with higher values of $R^*$. However, the majority of species (76%) still had negative Frescalo values at $R^* = 0.3$, compared to 80% at $R^* = 0.1$, and our analyses are concerned with relative change relating to species' climate and habitat associations. We also estimated species trends using the Telfer method[74], which is more conservative than Frescalo[12]. The Telfer approach calculates a metric of distribution change for each species relative to other species, calculating their deviation from the relationship of grid-cell occupancy in the historical versus the modern period. The method is thus based on the general assumption that rare species remain relatively rare and common species remain relatively common. Pearson correlation was lower between Frescalo ($R^* = 0.1$) and Telfer metrics (Pearson's rho = 0.51; Supplementary Fig. 2), although this increased to 0.78 using Spearman, indicating that the relative rank of each species' estimated change was similar across methods. Effects of using alternative values of species trends for analysis were also tested, see next section.

## Data analysis

Our first set of analyses aimed to examine how climate and habitat associations relate to regional distribution changes for the 1431 species for which climate and habitat associations were available. We built a linear mixed model using the lme4 package version 1.1.30[75], in which the dependent variable was the Frescalo distribution change metric, and the predictor variables were species mean temperature index, species temperature range index, grassland specialisation and forest specialisation. Interactions between climate and habitat associations were included, as well as random factors for province and species. Pre-analysis checks showed that collinearity between predictor variables was not an issue, with all variables having a variance inflation factor (VIF) lower than 2, and correlation scores were less than 0.7 (the highest being 0.62 between the two species climate indices)[76,77].

Predictor variables were standardised prior to analysis to allow interpretation of both single and interaction terms[78]. Significant effects were defined as those for which 95% confidence intervals did not include zero. Marginal and conditional $R^2$ were calculated as a measure of how well the dependent variable was explained by the fixed and fixed + random effects, respectively[79]. Models using alternative values of the Frescalo metric as the response variable produced very similar outputs as those discussed in the results, with species with warmer temperature associations, larger geographic ranges and higher forest specialisation values more likely to increase, while grassland specialists were more likely to decrease (Fig. 2, Supplementary Fig. 3, Supplementary Data 8). However, the negative interaction term between species temperature range index and forest specialisation was only significant in the original model with change metrics calculated with an $R^*$ value of 0.1 (Fig. 2c, Supplementary Fig. 2). In the alternative model where the Telfer distribution change metric was the response variable, the negative effect on grassland specialisation was no longer significant. However, a positive interaction between forest specialisation and species mean temperature index indicated that species that are associated with warm climates that are also forest species were especially likely to increase their regional distributions, in line with other findings in the study (Fig. 4b, Supplementary Fig. 3).

To assess the direction of regional distribution change in terms of climatic space, we used a subset of well-recorded cells, which we defined as those that contain observations of at least 10% of the province's observed species and at least 25% of the species found in the eight adjacent grid cells in each time period. This resulted in 813 grid cells for this analysis, which also passed the above checks for collinearity for environmental variables. Although such arbitrary cut-off points are generally not recommended for accounting for unknown recorder effort[12], it was in this case necessary to apply some measure through which we could consider that presences and absences in the raw data should be reliable for the subsequent calculations of each species' occupied climatic space[20,21]. For each species across all provinces in which it was present, we calculated the mean value of the mean annual temperature in occupied grid cells for both the historical and modern time period. Values were calculated for both historical and modern distributions according to the 1961–1990 reference period temperature, as well as for the historical distributions according to the 1961–1970 historical climate data, and for the modern distributions according to the 2001–2010 modern climate data. Shifts in climatic space were then calculated by subtracting the mean annual temperature of the historical distribution from that of the modern distribution. Paired, one-sided Wilcoxon tests were then carried out to assess whether the average temperatures of the modern range were cooler than those of the historical range according to the reference period temperature, which would indicate a climatic shift towards grid cells with cooler reference-period temperatures. This was based on the 1391 species that were observed in well-recorded grid squares for which gridded climate data were available. We also used a linear model to see whether these shifts could be explained by the climatic or habitat associations of the 1296 species for which habitat and climate associations were also available. The response variable was the shift in reference-period climate space, and as before, the predictor variables were species mean temperature index, species temperature range index, grassland specialisation and forest specialisation, as well as habitat-climate interactions. We also included controls for the Frescalo trend ($R^* = 0.1$) of each species (mean value across provinces for those species that were observed in more than one), as well as the number of provinces in which the species was recorded. Including alternative Frescalo or Telfer species trend values had no effect on the model outcome in terms of significant predictor variables (Supplementary Fig. 4, Supplementary Data 9).

To examine the effect of the grid cell environment to species turnover between the historical and modern time period, we created a two suites of generalised linear mixed models to establish how climate and land use explain extirpation (the fraction of species present in the historical time period that were not observed in the modern time period) and colonisation (the fraction of the province's species pool that was not present in the historical time period, but was observed in the modern time period). Each suite of models was then split into subsets of three, in which the environmental predictor variables were related to either the historical environment, modern environment, or the change over time. For habitat, the historical models contained historical grassland habitat, the modern models contained retained grassland habitat, and the change models contained grassland abandonment (see above). For climate, historical models contained mean annual temperature 1961–1970, modern models contained mean annual temperature 2001–2010, and change models contained the difference between historical and modern temperatures. Collinearity across predictor variables was again mild, with VIFs <2 and correlations = <0.6. Further, within each suite and subset of models, a model was created considering all 1701 species, and then for subgroups of species of interest, according to the results of earlier analyses. For extirpation, these were grassland specialists (grassland specialisation value of 5 or more $n = 347$), declining species (species with significantly negative Frescalo values[67]; Bohuslän $n = 555$, Öland $n = 130$, Uppland $n = 243$, Medelpad $n = 170$), and cool-associated species (species with species mean temperature index equal to or lower than the first quartile $n = 400$). For colonisation, these were forest specialists (forest specialisation value of 5 or more $n = 118$), expanding species (Bohuslän $n = 172$, Öland $n = 7$, Uppland $n = 50$, Medelpad $n = 18$), warm-associated species (species with species mean temperature index equal to or greater than the third quartile $n = 400$) and climatically widespread species (species with species temperature range index equal to or greater than the third quartile $n = 400$). In all there were 27 models, for which the number of grid cells included varied, depending on historical or modern observations of the subsets of species (Supplementary Data 6). The models were structured as follows. Extirpation or colonisation was the response variable, with habitat and climate and their interactions as predictors of interest. For the colonisation models in the forest specialist subgroup of species, grassland predictors were replaced by historical and modern forest cover (historical and modern conditions, respectively), while grassland abandonment (i.e. forest gain on former grassland) was retained for the change model. All models also included controls for sampling effort in each cell extracted from the Frescalo calculations (i.e. $1/\alpha_i$, the inverse of the sampling effort multiplier), the first two eigenvectors of a principal coordinates analysis derived from a neighbour matrix of the spatial coordinates of the centroid of each cell[80,81], the standard deviation of microclimate temperatures from a 50 m grid within each cell[82] and latitude. Because of the collinearity between latitude and temperature, we adopted a sequential regression approach to extract the residual variation after removal of latitude's effect on climate[77]. Residuals were extracted from separate linear models (one each for historical, modern and change) with each landscape's latitude as the dependent variable, and temperature as the predictor variable. Fixed predictor variables were standardised prior to analysis. A random effect was included for province, as well as an observation-level random effect to account for overdispersion in the model. Because the response variables were fractions, we used generalised linear mixed models with a binomial distribution, with model weights added to represent the number of 'trials'. For extirpation models, weights were the number of species observed in the cell in the historical period (i.e. those that could potentially be extirpated), while for colonisation models, weights were the number of species in the province in the historical period that were not present in the focal cell (i.e. those that could potentially colonise). Analyses of extirpation can be considered reliable, because they are based on observations in the historical period, and the consistently high observer effort in the modern period means that false absences are unlikely[30]. Colonisation estimates are more prone to error due to the reliance on true absences in the historical period, although we are confident that the Frescalo-derived estimates of sampling effort minimise the risk of potential bias in our models. Nonetheless, these results should be interpreted with reasonable caution. Significant effects were defined as those for which $p < 0.05$. Marginal and conditional $R^2$ were calculated using the performance package (version 0.9[83]). Figures were created with the help of the visreg (version 2.7[84]) and raster (version 3.6.2[85]) packages.

### Reporting summary

Further information on research design is available in the Nature Portfolio Reporting Summary linked to this article.

## Data availability

Processed datasets used for analysis are available in the Supplementary files (Supplementary Data 1, 3 and 5). For raw data, historical and modern species observations are available from the Swedish species gateway ArtPortalen (https://artportalen.se; historical observations from Öland and Bohuslän scheduled for upload December 2022). Species temperature indices were taken from https://doi.org/10.6084/m9.figshare.8845832.v1[66], while habitat specialisation information was extracted from the supplementary information of Tyler et al.[13]. Climate data are free to access from ftp://ftp.smhi.se/; contact kundtjanst@smhi.se for log-in details. Historical land-cover data are published at https://doi.org/10.17045/sthlmuni.4649854.v2[66], while modern land cover is published by the Swedish Environmental Protection Agency at https://metadatakatalogen.naturvardsverket.se/metadatakatalogen/GetMetaDataById?id=8853721d-a466-4c01-afcc-9eae57b17b39. The Swedish taxonomic database used for species name harmonisation is found at https://www.dyntaxa.se.

## Code availability

R code used to run analyses from the processed datasets (Supplementary Data 1, 3, 5) to produce model outputs (Supplementary Data 2, 4 and 6) are available in the Supplementary Software.

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

## Acknowledgements
This work would not have been possible without the historical, recent and continued dedication of the skilled botanists who spend their time recording plant species occurrences and producing fantastic provincial floras. Many thanks also to Andrew Suggitt for helpful comments on a previous version of the manuscript. A.G.A. is supported by the Swedish research councils Formas (2015-1065) and VR (2020-04276). J.-C.S. considers this work a contribution to his VILLUM Investigator project "Biodiversity Dynamics in a Changing World" funded by VILLUM FONDEN (grant 16549) and Center for Ecological Dynamics in a Novel Biosphere (ECONOVO), funded by Danish National Research Foundation.

## Author contributions
A.G.A. and J.-C.S. conceived and planned the work. A.G.A. collated data, performed the analysis and wrote the first draft. A.G.A. and J.-C.S. reviewed and edited subsequent drafts.

## Funding

## Competing interests
The authors declare no competing interests.
