## [Peer Review File · Nature Communications]

Review comments, first round -

Reviewer #1 (Remarks to the Author):

The authors assessed the interacting effects of changing climate and land use on the regional distributions of species by comparing historical and modern species inventories. The work focuses on four provinces in Sweden. Key findings include species from warmer climates expanding their range, grassland specialists showing decline related to managed grassland abandonment, and local extirpations of species related to climate warming can be reduced by grassland management.

The work is well done, timely, and fairly novel. I think it can be an important contribution that informs debate over the management of grassland biodiversity versus natural succession into forested ecosystems post-management abandonment. However, there are a few methodological and reporting details that I believe will strengthen the reader's confidence and understanding of the work that I detail below.

Major concerns:

I think it would be worth being more specific with "regional distribution" rather than "distribution" when referring to species' geographical distributions. It's a technical point, but the study is not about the species distributions in themselves but rather about how their distributions have shifted within the study regions.

A big concern with this dataset is the possibility of disparities in sampling effort and habitat coverage across time periods driving some patterns. The authors try to account for this with the Frescalo algorithm. However, the level of detail and information on how different sampling effort was between time periods and the Frescalo algorithm implementation leaves a lot of questions remaining. For instance, the Frescalo algorithm cannot get around if certain habitats were not sampled, such as early successional transition habitats. Were there any such disparities in particular habitats being sampled between time periods? Can the authors provide summary statistics about how different the sampling effort and data sources were between periods? How sensitive are the results to different Frescalo implementation parameters (e.g., different alpha levels)? Can extirpations and colonizations be explained by disparities in sampling efforts between time periods and certain habitats? I think the readers will have more confidence in the work with a supplement explaining more about the data and addressing the questions above.

Section on shifts in climatic space – Is the decision on lines 151-155 conflating cell diversity with sampling effort? Like a cell could be naturally low in diversity and not necessarily less well sampled. What are the implications of this for your results?

I think the choice to frame the shifts ("shifts in climatic space" section) in climatic space as historical climate minus modern is confusing. Different folks will have different opinions about this. However, making the reference point the historical climate is confusing because it requires more mental energy to say that they occupied cooler temperatures meaning warming has shifted their ranges warmer. Flipping it to modern minus historical streamlines the interpretation to "the species are occupying warmer sites than historical, reflecting a range shift with warming". It also streamlines interpretation with experienced temperature and makes intuitive the "counter-intuitive" result that cool distributed species are more vulnerable to climate warming. This has been shown experimentally in a number of studies (i.e., Reich et al. 2015 in Nature Climate Change, Liu et al. 2018 in New Phytologist, and Lynn et al. 2021 Global Change Biology). This same critique applies to Figure S1 and "Local drivers of turnover" results".

Minor comments:

Line 39- what is lagging here? Is this a prediction of a warming-caused extinction debt?

Line 65 – drop "actually do"

Line 76-77- try "changes to their habitat and climate preferences"

Line 78-79- Maybe just make it "we investigated how changes in the condition" – drop the parentheses.

Line 92- what is the scale of for the species occurrence maps? Also on a 5x5 km grid?
Line 94 – How was this determined (panel e)? It appears that most decline occurred in the middle (~6 deg. C) of their temperature range.
Line 102 – Not sure floral is the right word. Maybe “vegetation” or even “flora”?
Line 106-107 – “these estimates” being the probability a species occurs in the cell, correct? Maybe clarify a bit more. How are the probabilities compared across time periods? Is it probability of occurrence over recent years – probability of occurrence historically?
Line 110-112- is the mean temperature index historic temperature, present temperature, or both?
Line 120-122 – Maybe try “and extend the observation of local-scale extirpation of grassland species to an overall decline in their broad-scale distribution.”? Bit cleaner wording.
Line 123-125 – cool conclusion!
Line 128 or Figure 2- Would it be possible to report R^2 with this figure? It appears that the trends are significant (high sample size) but the explanatory power is low- can see that in the supplement. Good to be honest about it here.
Line 157- Be more explicit here to reduce confusion- say the difference is historical - modern.
Line 209- be careful with the wording here. It reads as mean temperature of current minus temperature of historic, such that negative values indicate species shifting towards cooler, rather than warmer ranges. Which is contrary to the interpretation and explanation in the results section.
Line 244 – add “lower the extirpation reducing effect of grassland retention”
Line 278- “deadly cocktail”? And is it habitat destruction? The grasslands are mostly abandoned from prior management (e.g., livestock grazing). In which case, it is natural secondary succession driving changes in the communities, not habitat destruction per se.
Line 286- Again, habitat destruction does not seem like the right framing. Destruction suggests that there some disturbance like deforestation or new agriculture performed on the system. What you mostly track is abandonment and natural succession in the Anthropocene.
Lines 346-350 – was this based only on modern observations or did it include historical?
Line 421-423- This reads like modern minus historic when you did historic minus modern.
Line 446 – should be ≤ 0.6

Reviewer #2 (Remarks to the Author):

- What are the noteworthy results?

The authors show that climate and grassland abandonment and their interaction are significant drivers of distributional change in plants species across a long time period and within a number of Swedish provinces. This is noteworthy.

- Will the work be of significance to the field and related fields? How does it compare to the established literature? If the work is not original, please provide relevant references.

Below I give one reference to a similar study that the authors ought to consider citing. I am not one of the authors I should add!

The work is original in time-scale of change and in the rigour and clarity of the analytical results, which support their conclusions. R^2 values are low having removed random effects but this is not unusual in these kinds of large, noisy multi-species analyses.

- Does the work support the conclusions and claims, or is additional evidence needed?

I think the work is rigorous. My one worry regards controlling for observer effort. They use Frescalo, an established technique, but it would have been useful to perform or cite some testing of the ability of Frescalo to handle differences in recorder effort within each survey.

- Are there any flaws in the data analysis, interpretation and conclusions? - Do these prohibit publication or require revision?

I can see no flaws but have my main recommendation would be for far greater clarity when describing some of the results. See my comments on the text below.

- Is the methodology sound? Does the work meet the expected standards in your field? Yes.
- Is there enough detail provided in the methods for the work to be reproduced? Yes.

Comments on text:

Lines 76-77: Probably more understandable if rephrased as follows "First, we quantify species' distribution changes in terms of their habitat and climatic preferences".

Lines 154-155: "to compare how species' distribution changes from the historical to the modern period have resulted in shifts in climate space". This is confusing. How can climate space change in response to species distribution changes? This needs rewording.

Lines 175-177: "Results indicate that warm-climate, southerly species are more likely to have shifted to historically warmer ranges, while the opposite dynamic is seen for cold-climate, northerly species (Figure 3b)." I think Fig 3b and the text need a clearer explanation. For example it seems to me that Fig 3b shows that species with preferences for a warmer climate (high values on X axis) have shifted to areas that have warmed relative to their historical climate i.e. a positive value when the lower historical temperature is subtracted from the higher and warmer modern temperature. What am I misunderstanding here? Explain more clearly thank you.

Line 178-179: "one might expect warm-related species to be more likely to expand to cooler areas" This is now thoroughly confusing. I presume they mean one might expect warm-related species are more likely to expand to historically cooler areas as their distribution tracks a warming climate?

Lines 182-184: "Second, relatively cold-climate species can over time occupy historically cooler ranges as they are gradually extirpated from the (relatively) warmer areas of their ranges 20. Such extirpations could be due to temperatures becoming unsuitably high" Again, very confusing. Here the phrase "historically cooler" implies these are now warmer in modern times so why would cold-climate species move into these areas!?

Lines 187-191 "Remembering from the first analysis that distributions of grassland species are declining as a whole, faster declines in warmer areas can potentially be explained by likely faster levels of secondary succession following grassland abandonment 7,21. This is supported by a significant interaction term, showing that southerly-distributed grassland specialists are 'shifting' to cooler ranges at a faster rate" This interpretation is key but I don't think the interaction term proves the mechanism of faster secondary succession following grassland abandonment. For example have grasslands reduced more greatly in extent in the warmer areas hence land-use change is a stronger driver both directly but also perhaps because of extinction debt?

Lastly the following reference will be of interest and relevance and ought to be cited.

Reference:

Hill, MO & Preston CD (2015) Disappearance of Boreal plants in southern Britain: habitat loss or climate change? *Biol.J.Linn.Soc* 115, 598-610.

Simon Smart

Response to referees

Thank you for your patience while your manuscript "Climate warming compounds plant responses to habitat conversion" was under review. We have now received reports from two referees and, on the basis of their comments, we have decided to invite a revision of your work for further consideration in our journal. Your revision should address all the points raised by our reviewers (see their reports below). In particular, please note the reviewers' concerns on potential sources of bias in the dataset and the estimation of shifts in climate space. We also ask for more transparency in reporting the R-squared values in the main text and/or figures. Please note that there is no word limit on the Methods section, so please include as much detail as necessary to explain your work and to fulfil the queries of the reviewers

Thank you for considering a revised version of our manuscript. We have worked hard to convince the reader regarding uneven observer effort and the choices that we made for dealing with it. This includes a more thorough description of both the dataset and the Frescalo method, as well as some analyses for assessing the sensitivity of our results to different input parameters for Frescalo, plus the use of another, more conservative metric for estimating distribution change. We have also re-written other parts of the manuscript to improve clarity, especially regarding species' shifts in climatic space. We have added and commented on R^2 values in the main manuscript, and made other minor changes in response to the referee comments. Full details are found in our responses to the individual reviewer comments.

Alistair Auffret & Jens-Christian Svenning

REVIEWER COMMENTS

Reviewer #1 (Remarks to the Author):

The authors assessed the interacting effects of changing climate and land use on the regional distributions of species by comparing historical and modern species inventories. The work focuses on four provinces in Sweden. Key findings include species from warmer climates expanding their range, grassland specialists showing decline related to managed grassland abandonment, and local extirpations of species related to climate warming can be reduced by grassland management.

The work is well done, timely, and fairly novel. I think it can be an important contribution that informs debate over the management of grassland biodiversity versus natural succession into forested ecosystems post-management abandonment. However, there are a few methodological and reporting details that I believe will strengthen the reader's confidence and understanding of the work that I detail below.

Thanks for your review

Major concerns:

I think it would be worth being more specific with "regional distribution" rather than "distribution" when referring to species' geographical distributions. It's a technical point, but the study is not about the species distributions in themselves but rather about how their distributions have shifted within the study regions.

We have now replaced 'distribution' with 'regional distribution' where appropriate (e.g. Lines 33, 71, 76, 102 and throughout).

A big concern with this dataset is the possibility of disparities in sampling effort and habitat coverage across time periods driving some patterns. The authors try to account for this with the Frescalo algorithm. However, the level of detail and information on how different sampling effort was between time periods and the Frescalo algorithm implementation leaves a lot of questions remaining. For instance, the Frescalo algorithm cannot get around if certain habitats were not sampled, such as early successional transition habitats. Were there any such disparities in particular habitats being sampled between time periods? Can the authors provide summary statistics about how different the sampling effort and data sources were between periods? How sensitive are the results to different Frescalo implementation parameters (e.g., different alpha levels)? Can extirpations and colonizations be explained by disparities in sampling efforts between time periods and certain habitats? I think the readers will have more confidence in the work with a supplement explaining more about the data and addressing the questions above.

Thank you for this comment. Of course, we cannot know exactly how observer effort has varied in time and space, but we do know that it is important to consider this variation and to try to account for it. We have made a number of changes to the manuscript to convince readers that our data are of sufficient quality for analysis and that the Frescalo algorithm (and our choices regarding parameters) is appropriate for correcting for observer effort.

First, we provide more information on the data collection for the historical and the modern floras (Lines 420-440). We argue that despite consistently higher sampling effort in the modern floras (as broadly measured by the number of unique species-grid cell observations), the aim of both the historical and modern floras to characterise distributions of the species in the province means that our data are sufficiently comprehensive to both assess species-level distribution change and grid-cell level turnover. Specifically, even in the lower-intensity historical period, 40-61% of all observations were species-grid cell duplicates, while the species covered represent a spread in grassland and forest specialisation values. Together, this gives a strong indication that provinces were broadly well-visited, and that surveys covered a broad range of habitats. Nonetheless, it is important to try to correct for any bias in observer effort.

We chose Frescalo following Isaac et al. (2014; <https://doi.org/10.1111/2041-210X.12254>), who compare several methods, and recommend Frescalo for datasets such as biodiversity atlases that report species presences (rather than abundances) across discrete time periods (rather than time series). We now mention this in both the main text (Lines 111-112) and the methods, as well as citing examples of atlases, research publications and policy reports that use the approach (Lines 454-458), to add further confidence for the reader. As an aside (but not suitable for the manuscript), author Alistair Auffret has worked with the botanical society of one of the provinces (Öland), and the Frescalo change values over time (including an intermediate time period not part of this study) were approved for publication in the upcoming book by the core flora editorial group after careful consideration of the outputs compared to their expert knowledge. In the methods, we now provide much more detail regarding the Frescalo approach (Lines 443-454).

In the original manuscript, rather than accept all default values for parameters in the Frescalo algorithm, we tried to apply relevant values based on our data. Specifically, we changed the value of the fraction of most common species used to define benchmark species for change metric calculation (now referred to as R^* , as in Hill 2011; <https://doi.org/10.1111/j.2041-210X.2011.00146.x>, previously referred to in the manuscript as alpha, as in the sparta package, unfortunately because in Hill 2011 α refers to the sampling effort multiplier). The default R^* value is 0.27, but because two of our provinces have historical data based on maps which do not cover all species (in particular not including

the most ubiquitous species, for which a distribution map would not be useful), we chose the lower value of 0.1 to ensure that only the most common species could be used as benchmark species. We have now explained this choice more clearly in the methods (Lines 464-468). Furthermore, we recalculated species change metrics using R^* values of 0.2 and 0.3. These were strongly (0.98-0.99) correlated with the metrics that we originally calculated. Re-running our 'correlates of regional distribution change' with these values of distribution change as response variables provided very similar results (one interaction effect was no longer significant) that did not alter the message of our paper. Including these alternative change metrics as control variables in the 'shifts in climatic space' analyses gave identical results to the original analysis (Lines 470-485, 501-513, 541-543).

As a further step to support the robustness of the Frescalo method, in particular that it was not 'too clever' and providing spurious results, we also calculated an alternative distribution change metric. The Telfer approach (Telfer et al. 2002; [https://doi.org/10.1016/S0006-3207\(02\)00050-2](https://doi.org/10.1016/S0006-3207(02)00050-2)) estimates relative range change by calculating the deviation of each species from the regression line relating occupancy in two time periods, under the broad assumption that relatively rare species remain relatively rare, and relatively common species remain relatively common. This was found by Isaac et al (2014) to be a robust, but somewhat conservative measure of distribution change. Here, the Telfer metric was broadly correlated with our original Frescalo metric (0.51, but rising to 0.78 using Spearman). Again, replacing Frescalo change in our 'correlates of regional distribution change' found broadly the same results as the original analysis. The negative relationship between grassland specialisation and distribution change was no longer significant, while a positive interaction between the effect of species temperature association and forest specialisation was significant. The loss of a significant effect could be interpreted as Telfer being more conservative and not picking up on smaller changes in distributions (see also results in 'Local drivers of turnover' analysis identifying climate as a more prominent driver of floristic change compared to grassland loss), while the new interaction effect that was not found in the Frescalo-based models clearly matches our other results that suggest a reinforcing effect of climate warming and increased forest cover (Lines 470-485, 501-513, 541-543).

Because sampling effort is calculated in Frescalo before the consideration of benchmark species, we did not add any auxiliary analyses to the final 'Local drivers of turnover' analysis. However, we believe that our added detail and citations, and the agreement between Frescalo and conservative Telfer change metric (and small influence on other results) presents a strong case for including the Frescalo sampling effort estimate. We also believe that we can be particularly confident in the analysis of extirpations according to environmental variables. This is because the species status relies heavily on the modern period, which was characterised by a consistently high sampling effort across provinces. Therefore, if a species was present in the modern period, then it was very likely observed, and so false extirpations (present in historical, absent in modern period) and false persistences (present in historical and modern periods) are probably very rare. Indeed, including only persistence/extirpation as a response can be a way to analyse species change where historical data are especially patchy (e.g. Suggitt et al. 2018; <https://doi.org/10.1038/s41558-018-0231-9>). Colonisations are in general more prone to error, being reliant on true absences in the historical period. However, as explained above, we believe that the historical data here, together with Frescalo estimates of observer effort, mean that we can be confident in our results. Nonetheless, we have cautioned the reader with regards to this (Lines 588-594).

Finally, we are satisfied that the results are consistent both across our different analyses, as well as what we know about grassland loss, climate change and how we would expect them to affect plant species.

Section on shifts in climatic space – Is the decision on lines 151-155 conflating cell diversity with sampling effort? Like a cell could be naturally low in diversity and not necessarily less well sampled. What are the implications of this for your results?

This is a good point and an example of why, where possible, one should avoid arbitrary cut-offs with regards to sampling effort (see again Isaac et al. 2004). We now tell the reader why we need to do this, in the main text (Lines 160-162) and the methods (Lines 519-523). We also cite examples of other studies that have found it necessary to take such an approach when estimating species' range shifts. Although the point raised here is valid, we think that 25% of the surrounding diversity, plus 10% of the regional species pool is quite a low water mark for inclusion, and we do retain 2/3 of our grid cells. To have any considerable effect on these results, any cells that are low in diversity but still well-sampled would have to be both somewhat over-represented in the cells that species colonise or are extirpated from over time (to affect change in occupancy), and be close enough to the extremes of their climatic ranges (for changes in occupancy to affect calculated climate space). We think that this is unlikely, and as above must take comfort from the fact that our results make sense in relation to the other results in the study, for which we are able to take a more robust approach to unknown sampling effort.

I think the choice to frame the shifts (“shifts in climatic space” section) in climatic space as historical climate minus modern is confusing. Different folks will have different opinions about this. However, making the reference point the historical climate is confusing because it requires more mental energy to say that they occupied cooler temperatures meaning warming has shifted their ranges warmer. Flipping it to modern minus historical streamlines the interpretation to “the species are occupying warmer sites than historical, reflecting a range shift with warming”. It also streamlines interpretation with experienced temperature and makes intuitive the “counter-intuitive” result that cool distributed species are more vulnerable to climate warming. This has been shown experimentally in a number of studies (i.e., Reich et al. 2015 in *Nature Climate Change*, Liu et al. 2018 in *New Phytologist*, and Lynn et al. 2021 *Global Change Biology*). This same critique applies to Figure S1 and “Local drivers of turnover” results”.

With regard to this, later comments by the same referee and several comments from the other referee, we regret that our explanation in this part of the main manuscript was not of the standard that we would have liked.

We have now carefully re-written this part of the manuscript, guiding the reader more carefully through the calculation process and what the results mean (Lines 151-213). Unfortunately, in the original manuscript we used the word ‘historical’ in a confusing way. Now, we explain more clearly that overall ‘shifts to cooler ranges’, both overall and later specifically for e.g. cold-climate species, relates to shifts in space along a geographic climate gradient. That is, we use reference period (1961-1990) temperatures for all grid cells (so southerly grid cells generally have higher, and northerly grid cells lower temperatures), and compare occupancy in the historical period and the modern period according to these reference temperatures. Therefore, we do subtract the mean reference period temperature across a species' historical distribution from the mean reference period temperature across the modern range to find the change in climate space. As such, a negative value (higher average temperature in the historical period, lower average temperature in the modern period) refers to a species that has shifted more or less northward to a relatively cooler reference-period climate. Note that we do also calculate the shift in experienced climate, whereby all species are now experiencing warmer ranges due to the climate change that has occurred across the study regions, but we think that this part is generally clear, and these

results are not used in any analytical models for which parameter estimates need to be interpreted.

Throughout this section, we are now clear in every sentence as to which time periods (historical or modern observations; historical, modern or reference period temperatures) are used when, and how. We have also added several examples during the explanation, to spell out what a positive or negative value means in terms of species responses. We hope that this has made the section easier to follow, and that our ecological explanations of our results now make more sense. Many thanks for the suggested papers above, we now cite two of them to support our finding that cold-climate species appear to be retracting their climatic distributions.

Minor comments:

Line 39- what is lagging here? Is this a prediction of a warming-caused extinction debt? Yes, because climatic niche theory predicts that species' ranges are determined by climate, the strong increase in the temperatures experienced by temperatures across their range over time indicates that future local extinctions should be expected. We have added this briefly to the abstract (Lines 39-40).

Line 65 – drop “actually do”
Changed as suggested (Line 66).

Line 76-77- try “changes to their habitat and climate preferences”
We have changed this in line with the suggestion of referee 2: “First, we quantify species' distribution changes in terms of their habitat and climatic preferences” (Lines 77-78).

Line 78-79- Maybe just make it “we investigated how changes in the condition” – drop the parentheses.
Changed to: Then, we study how changes in the environmental conditions in the landscape affect the level of community turnover that has occurred over time (Lines 79-80).

Line 92- what is the scale of for the species occurrence maps? Also on a 5x5 km grid?
Species occurrence resolutions vary according to time period and province. but 5x5 km is the smallest size that we can confidently assign the historical observations to (as well as it matching the inventory unit for modern floras). We have changed the figure text to read: “Observation data were taken from historical and modern plant atlases, translated into occupancy in 5 x 5 km grid cells” (Lines 92-93).

Line 94 – How was this determined (panel e)? It appears that most decline occurred in the middle (~6 deg. C) of their temperature range.
The lower panels of this figure were mainly included to give a visual representation of the dataset using an example species. We chose *Crepis praemorsa* because our findings of negative trends are supported by its presence on the expert-derived red list. However, the occupancy map and histogram are based only on the raw observations and we shouldn't read too much into this. Therefore, we have changed the wording to specify both that we are looking at recorded observations, as well as being more general in terms of the change in terms of climate. It now reads: “Modern observations appear to be especially fewer in the southerly, warmer half of its distribution” (Lines 95-96).

Line 102 – Not sure floral is the right word. Maybe “vegetation” or even “flora”?
We have now changed to ‘floristic similarity’, which seems to be a term that gets used in the literature (Line 106).

Line 106-107 – “these estimates” being the probability a species occurs in the cell, correct? Maybe clarify a bit more. How are the probabilities compared across time periods? Is it probability of occurrence over recent years – probability of occurrence historically?

We have now added more information on the Frescalo calculation both here (Lines 105-112), as well as in our much more detailed description in the methods (Lines 443-454).

Line 110-112- is the mean temperature index historic temperature, present temperature, or both?

This information was given in the methods (Lines 379-384), but we have now added more detail in the main text (Lines 116-119). Species’ temperature indices are calculated from available observation data from provincial floras covering Sweden (not just these four provinces) published since 1975, with temperatures for each occurrence calculated according to both time and place of each observation.

Line 120-122 – Maybe try “and extend the observation of local-scale extirpation of grassland species to an overall decline in their broad-scale distribution.”? Bit cleaner wording.

Changed as suggested (Lines 126-129).

Line 123-125 – cool conclusion!

Indeed. It’s a good time to be a forest generalist plant in Sweden right now, at least compared to grassland specialists!

Line 128 or Figure 2- Would it be possible to report R^2 with this figure? It appears that the trends are significant (high sample size) but the explanatory power is low- can see that in the supplement. Good to be honest about it here.

We have added R^2 values to the figure text of each figure (Figs 2-4), and commented on our overall low values in the discussion (Lines 319-326).

Line 157- Be more explicit here to reduce confusion- say the difference is historical - modern.

Line 209- be careful with the wording here. It reads as mean temperature of current minus temperature of historic, such that negative values indicate species shifting towards cooler, rather than warmer ranges. Which is contrary to the interpretation and explanation in the results section.

These relate to the confusion stemming from our original description of this analysis. Please see our response to the initial comment above.

Line 244 – add “lower the extirpation reducing effect of grassland retention”

Changed as suggested. The point is much clearer now (Line 267).

Line 278- “deadly cocktail”? And is it habitat destruction? The grasslands are mostly abandoned from prior management (e.g., livestock grazing). In which case, it is natural secondary succession driving changes in the communities, not habitat destruction per se.

Line 286- Again, habitat destruction does not seem like the right framing. Destruction suggests that there some disturbance like deforestation or new agriculture performed on the system. What you mostly track is abandonment and natural succession in the Anthropocene.

Yes, in this case we refer to the so-called deadly cocktail of habitat destruction and climate change. Although we do not wish to start a debate, we do believe that natural systems do/did include grazing dynamics that are not present in pastures that are abandoned today. However, we understand that readers might not instinctively see what we focus on here as habitat destruction. Therefore, we have added a qualifier that in this case we consider grassland abandonment to be habitat destruction, including some references such as the

European Red List, that names the grazing abandonment several times as a major threat for plant species and populations (Lines 303-304).

Lines 346-350 – was this based only on modern observations or did it include historical?
This comment regarding the calculation of species climate indices is answered above in response to a similar comment for the main text.

Line 421-423- This reads like modern minus historic when you did historic minus modern.
This is another result of the confusion regarding shifts in climatic space, which we hope are now clear.

Line 446 – should be ≤ 0.6
Changed (Line 556).

Thanks again for your comments.

Reviewer #2 (Remarks to the Author):

- What are the noteworthy results?

The authors show that climate and grassland abandonment and their interaction are significant drivers of distributional change in plants species across a long time period and within a number of Swedish provinces. This is noteworthy.

Thank you

- Will the work be of significance to the field and related fields? How does it compare to the established literature? If the work is not original, please provide relevant references.

Below I give one reference to a similar study that the authors ought to consider citing. I am not one of the authors I should add! The work is original in time-scale of change and in the rigour and clarity of the analytical results, which support their conclusions. R2 values are low having removed random effects but this is not unusual in these kinds of large, noisy multi-species analyses.

Thanks. We have now cited the suggested paper, which fits well as an additional example of species declining at their warmer range edges (Line 202). R2 values have been added to all figure texts (Figs 2-4), and we have commented on the low values in the discussion (Lines 319-326).

- Does the work support the conclusions and claims, or is additional evidence needed?

I think the work is rigorous. My one worry regards controlling for observer effort. They use Frescalo, an established technique, but it would have been useful to perform or cite some testing of the ability of Frescalo to handle differences in recorder effort within each survey.

As with our response to referee 1, we cannot know for sure about observer effort in the past, and how adequately Frescalo accounts for it. Nonetheless, we hope that our additions in the text, plus additional analyses including alternative outputs of Frescalo and the more conservative Telfer metric, will give readers greater confidence in our approach and the results that were produced. Please see extended response to the previous referee's comment for details.

- Are there any flaws in the data analysis, interpretation and conclusions? - Do these prohibit publication or require revision?

I can see no flaws but have my main recommendation would be for far greater clarity when describing some of the results. See my comments on the text below.

Thank you. We respond to those specific comments below.

- Is the methodology sound? Does the work meet the expected standards in your field?

Yes.

Thank you

- Is there enough detail provided in the methods for the work to be reproduced?

Yes.

Thank you.

Comments on text:

Lines 76-77: Probably more understandable if rephrased as follows "First, we quantify species' distribution changes in terms of their habitat and climatic preferences".

We have altered this sentence as suggested (Lines 77-78).

Lines 154-155: "to compare how species' distribution changes from the historical to the modern period have resulted in shifts in climate space". This is confusing. How can climate space change in response to species distribution changes? This needs rewording.

We have now re-worded to make it clear that what changes is the climate space that each species occupies, rather than the climate itself: "To assess the direction of regional distribution gains and losses in terms of climatic ranges, a second set of analyses compared how species' regional distribution changes from the historical to the modern period have resulted in shifts in the climate space that they occupy." (Lines 152-154).

Lines 175-177: "Results indicate that warm-climate, southerly species are more likely to have shifted to historically warmer ranges, while the opposite dynamic is seen for cold-climate, northerly species (Figure 3b)." I think Fig 3b and the text need a clearer explanation. For example it seems to me that Fig 3b shows that species with preferences for a warmer climate (high values on X axis) have shifted to areas that have warmed relative to their historical climate i.e. a positive value when the lower historical temperature is subtracted from the higher and warmer modern temperature. What am I misunderstanding here? Explain more clearly thank you.

As with the confusion caused for the other referee, we do regret not explaining this better in the original version of the manuscript. Please see the response to their initial comment, and the changes in the manuscript for more clarity. Specifically for this comment, Figure 3b shows that species with cooler climatic associations have retracted to occupy grid cells that have on average lower values according to the 1961-1990 reference period (~more northerly areas), in line with our interpretation that cool-related species are being extirpated from their warmer range edges. On the other hand, warm-related species are occupying on grid cells that have on average higher temperature values in the 1961-1990 reference period, in line with our interpretation that they are consolidating their existing range ('range filling') to a greater extent than they are expanding their range edge.

Line 178-179: “one might expect warm-related species to be more likely to expand to cooler areas” This is now thoroughly confusing. I presume they mean one might expect warm-related species are more likely to expand to historically cooler areas as their distribution tracks a warming climate?

Yes, your interpretation is correct. We have now clarified: “one might expect warm-related species to be the species that are expanding to higher latitudes, i.e. to grid cells with lower reference-period temperatures” (Lines 194-196).

Lines 182-184: “Second, relatively cold-climate species can over time occupy historically cooler ranges as they are gradually extirpated from the (relatively) warmer areas of their ranges 20. Such extirpations could be due to temperatures becoming unsuitably high” Again, very confusing. Here the phrase “historically cooler” implies these are now warmer in modern times so why would cold-climate species move into these areas!?

Again, we apologise for the confusion. We hope that the improved section is clearer.

Lines 187-191 “Remembering from the first analysis that distributions of grassland species are declining as a whole, faster declines in warmer areas can potentially be explained by likely faster levels of secondary succession following grassland abandonment 7,21. This is supported by a significant interaction term, showing that southerly-distributed grassland specialists are ‘shifting’ to cooler ranges at a faster rate” This interpretation is key but I don’t think the interaction term proves the mechanism of faster secondary succession following grassland abandonment. For example have grasslands reduced more greatly in extent in the warmer areas hence land-use change is a stronger driver both directly but also perhaps because of extinction debt?

We are quite confident in our interpretation, as it does follow from existing knowledge regarding increased levels of densification in warmer climates, while our later analysis also supports reinforcing effects of climate change and grassland abandonment, and that we have low correlation in grassland loss and climate in our pre-analysis tests. Nonetheless, there may be other, additional factors lying behind this pattern (as mentioned above, we have quite low R^2 values), and we now mention that there could be other pressures affecting grassland species in warmer regions (Lines 212-213).

Lastly the following reference will be of interest and relevance and ought to be cited.

Reference:

Hill, MO & Preston CD (2015) Disappearance of Boreal plants in southern Britain: habitat loss or climate change? *Biol.J.Linn.Soc* 115, 598-610.

We have now cited this, as mentioned above.

Simon Smart

Many thanks again for your comments.

Additional minor changes

In filling out the reporting summary, we realised that we had only written the total of 1701 species in the text, whereas some of the analyses were based on fewer species because trait data were not always available. So, we have added this information to the text (Lines 488, 532, 534).

Another small change is that the Y-axis in Figure 1 had been standardized by the visreg package without us noticing (because the Frescalo trends are generally bunched around zero anyway). This has now been rectified, and the issue did not affect any of the other figures.

Review comments, second round -

Reviewer #1 (Remarks to the Author):

I was one of the original reviewers of the manuscript and I commend the authors for their thorough and well-performed revision of the manuscript. The authors have strongly clarified the writing around species temperature change over time and space. The additional analyses around the Frescalo method and other analyses with the Telfer algorithm ensure that the authors are presenting the most transparent view possible of the data, which will reassure readers of the robustness of the results. I have only a few minor comments (and a continued quibble that can be ignored but needs to be written) detailed below.

Minor comments:

Lines 39 – 40 – I still find this wording a little confusing. Maybe try “Strong temperature increases were experienced by species across their ranges but responses in their ranges appear time-lagged (e.g., lagging climate-driven local extinctions).”

Line 121 – Add between? i.e., “the difference between the warmest... ”

Line 167 – add “which resulted in...”

Line 304 – I appreciate the addition by the authors to bring more nuance to this point. Reasonable folks will disagree, but for me and a lot of other folks, “habitat destruction” means disturbance and destruction/removal of biomass. Abandonment and climate change are serious threats to biodiversity but do not fit into the category of “destruction”. The authors hint that they also think of climate change as different from habitat destruction in the following paragraph (lines 314-318) where they write that climate change effects are “in contrast” to habitat destruction. An alternative would be environmental or habitat change. I appreciate that the authors disagree and I do not recommend any further edits if they choose. I just want to bring this point forward and note the risk of terms losing their usefulness by becoming more opaque, general, and losing their referent.

Line 457 – remove “in”

Reviewer #2 (Remarks to the Author):

I am very impressed and satisfied with the way the authors have addressed my previous concerns. While ideally it would still give confidence to be able to directly model change in recorder effort over time I appreciate that this is an impossible request to level not just at these data but to many such analyses whose value remains high. The authors have made strenuous efforts to validate and interpret their results and so I am happy with the revision.

My one comment is rather trivial but in their References is an incomplete citation to Atlas 2020. This is not published yet and so it would be better to refer to it as in prep. It should be out in spring 2023.

REVIEWERS' COMMENTS

Reviewer #1 (Remarks to the Author):

I was one of the original reviewers of the manuscript and I commend the authors for their thorough and well-performed revision of the manuscript. The authors have strongly clarified the writing around species temperature change over time and space. The additional analyses around the Frescalo method and other analyses with the Telfer algorithm ensure that the authors are presenting the most transparent view possible of the data, which will reassure readers of the robustness of the results. I have only a few minor comments (and a continued quibble that can be ignored but needs to be written) detailed below.

Minor comments:

Lines 39 – 40 – I still find this wording a little confusing. Maybe try “Strong temperature increases were experienced by species across their ranges but responses in their ranges appear time-lagged (e.g., lagging climate-driven local extinctions).”

Now changed to: “Strong temperature increases were experienced by species across their ranges, indicating time lags in expected warming-related local extinctions”. This was inspired by the suggestion, but edited for length (Lines 38-39).

Line 121 – Add between? i.e., “the difference between the warmest... “

Added, thank you (Line 109).

Line 167 – add “which resulted in...”

Added, thank you (Line 139).

Line 304 – I appreciate the addition by the authors to bring more nuance to this point. Reasonable folks will disagree, but for me and a lot of other folks, “habitat destruction” means disturbance and destruction/removal of biomass. Abandonment and climate change are serious threats to biodiversity but do not fit into the category of “destruction”. The authors hint that they also think of climate change as different from habitat destruction in the following paragraph (lines 314-318) where they write that climate change effects are “in contrast” to habitat destruction. An alternative would be environmental or habitat change. I appreciate that the authors disagree and I do not recommend any further edits if they choose. I just want to bring this point forward and note the risk of terms losing their usefulness by becoming more opaque, general, and losing their referent.

Many thanks for this comment, and showing that researchers can respectfully disagree with one another. We have now changed ‘habitat destruction’ to ‘habitat conversion’ in these two instances, for a slightly softer and hopefully more widely acceptable term, which also matches with our title (Lines 236, 246).

Line 457 – remove “in”

Removed, thank you (Line 389).

Reviewer #2 (Remarks to the Author):

I am very impressed and satisfied with the way the authors have addressed my previous concerns. While ideally it would still give confidence to be able to directly model change in recorder effort over time I appreciate that this is an impossible request to level not just at these data but to many such analyses whose value remains high. The authors have made strenuous efforts to validate and interpret their results and so I am happy with the revision.

My one comment is rather trivial but in their References is an incomplete citation to Atlas 2020. This is not published yet and so it would be better to refer to it as in prep. It should be out in spring 2023.

We have now written the date of expected publication (21/03/2023) in the reference list (Line 689; see <https://press.princeton.edu/books/hardcover/9780691247595/plant-atlas-2020>).